# Clinical Utility of Liquid Biopsy to Detect BRAF and NRAS Mutations in Stage III/IV Melanoma Patients by Using Real-Time PCR

**DOI:** 10.3390/cancers14133053

**Published:** 2022-06-21

**Authors:** Emilio Francesco Giunta, Vincenzo De Falco, Pietro Paolo Vitiello, Luigi Pio Guerrera, Gabriella Suarato, Rossella Napolitano, Alessandra Perrone, Giuseppe Argenziano, Renato Franco, Michele Caraglia, Erika Martinelli, Davide Ciardiello, Fortunato Ciardiello, Stefania Napolitano, Teresa Troiani

**Affiliations:** 1Medical Oncology Unit, Department of Precision Medicine, Università degli Studi della Campania “Luigi Vanvitelli”, 80131 Naples, Italy; emiliofrancesco.giunta@unicampania.it (E.F.G.); vincenzodefalc@gmail.com (V.D.F.); luigipioguerrera@hotmail.it (L.P.G.); gabriella.suarato@gmail.com (G.S.); rossella.napolitano@unicampania.it (R.N.); alessandra.perrone@unicampania.it (A.P.); erika.martinelli@unicampania.it (E.M.); davideciardiello@yahoo.it (D.C.); fortunato.ciardiello@unicampania.it (F.C.); 2Candiolo Cancer Institute, FPO-IRCCS, 10060 Candiolo, TO, Italy; pietropaolo.vitiello@gmail.com; 3Department of Oncology, University of Torino, 10060 Candiolo, TO, Italy; 4Medical Oncology Unit, Ospedale Casa Sollievo della Sofferenza, 71013 San Giovanni Rotondo, Italy; 5Dermatology Unit, Department of Mental and Physical Health and Preventive Medicine, Università degli Studi della Campania “Luigi Vanvitelli”, 80131 Naples, Italy; giuseppe.argenziano@unicampania.it; 6Pathology Unit, Department of Mental and Physical Health and Preventive Medicine, Università degli Studi della Campania “Luigi Vanvitelli”, 80138 Naples, Italy; renato.franco@unicampania.it; 7Department of Precision Medicine, AOU Policlinico Vanvitelli, Università degli Studi della Campania “Luigi Vanvitelli”, 80138 Naples, Italy; michele.caraglia@unicampania.it

**Keywords:** liquid biopsy, polymerase chain reaction, melanoma, BRAF mutation, immunotherapy, targeted therapy

## Abstract

**Simple Summary:**

Liquid biopsy is an increasingly used tool for melanoma diagnosis and molecular characterization, but also for monitoring of response to anticancer drugs. The aim of our work is to assess the clinical utility of a real-time quantitative PCR (qPCR)-based platform with a very short turnaround time and identify the best setting for clinical investigation. We investigated the concordance of this technique with tissue analysis in stage III–IV melanoma patients; moreover, we correlated results to clinicopathologic characteristics and outcomes. We found a higher tissue–plasma concordance in melanoma patients with high burden of disease (sum of diameters ≥30 mm, ≥2 metastatic sites, elevated LDH levels), constituting a clinical subgroup worthy of future prospective evaluation; however, the low sensitivity of this technique seems to be not sufficient for predicting relapses in radically resected patients.

**Abstract:**

Background: Liquid biopsy is a potentially useful tool for melanoma patients, also for detecting BRAS/NRAS mutations, even if the tissue analysis remains the current standard. Methods: In this work, we tested ctDNA on plasma samples from 56 BRAF-V600/NRAS mutant stage III/IV melanoma patients using a real-time quantitative PCR (qPCR)-based platform. The study population was divided into two cohorts: the first including 26 patients who had undergone radical resection (resected cohort) and the second including 30 patients who had unresected measurable disease (advanced cohort). Moreover, for 10 patients in the advanced cohort, ctDNA assessment was repeated at specified timepoints after baseline testing. Data were analyzed and correlated to the clinicopathologic characteristics and outcomes. Results: In the baseline cohort, a higher tissue–plasma concordance was seen in patients with high burden of disease (sum of diameters ≥30 mm, ≥2 metastatic sites, elevated LDH levels); furthermore, monitoring of these patients through ctDNA analysis was informative for therapeutic responses. On the other hand, the low sensitivity of this technique did not allow for clinically valuable prediction of relapses in radically resected stage III/IV patients. Conclusions: Overall, our data suggest that qPCR-based ctDNA analysis could be informative in a subset of locally advanced and metastatic melanoma patients with specific clinical–radiological characteristics, supporting further investigations in this setting.

## 1. Introduction

Cutaneous melanoma is the most lethal type of skin cancer despite treatment advances. In fact, a high percentage of patients relapse after radical resection [1] and less than one patient out of two survive at 5 years from diagnosis of advanced disease [2]. BRAF and NRAS genes are frequently mutated in cutaneous melanoma, approximatively in about 50% and 20% of cases, respectively [3,4]. Guidelines recommend BRAF V600 and NRAS mutation testing on tumor specimens for resectable or unresectable stage III/IV melanoma [5]. However, intra-tumor heterogeneity is a well-established biological characteristic of human malignancies, including melanoma, affecting the predictiveness of tissue BRAF/NRAS mutational testing [6,7].

Liquid biopsy allows isolation of circulating tumor cells (CTCs) or circulating tumor DNA (ctDNA) in human samples [8] to detect tumor traces released from primary tumor and/or metastatic sites and to overcome tumor heterogeneity [9]. In particular, several studies conducted on metastatic melanoma patients have highlighted the utility of liquid biopsy in detecting and monitoring BRAF/NRAS mutations through the use of different technologies [10,11,12,13,14,15,16,17]. Some clinical and radiological characteristics have been correlated to ctDNA levels in metastatic melanoma patients, such as the LDH status (normal/high) and dimensional evaluation of metastatic lesions [15,18], whereas, in radically resected patients, ctDNA detection has prognostic significance, being associated to a higher relapse risk [19,20].

In this study, we evaluate the clinical utility of liquid biopsy for testing common BRAF/NRAS mutations using Idylla™ Biocartis, a fully automated real-time quantitative PCR (qPCR)-based platform, in 56 consecutive stage III–IV melanoma patients, divided in two main groups: 30 patients with locally advanced and metastatic melanoma, of which 25 were considered the baseline cohort because the liquid biopsy was collected before any systemic treatment (treatment-naïve), whereas for 5 patients the samples were performed on treatment (non-baseline); the remaining 26 patients had radically resected stage III–IV melanoma and blood samples were collected at baseline (treatment-naïve)

## 2. Materials and Methods

### 2.1. Patients’ Cohorts and Clinical Evaluation

In total, 56 consecutive BRAF-V600/NRAS mutant stage III/IV melanoma patients were assessed between September 2018 and April 2020 at the Oncology Unit of University of Campania “Luigi Vanvitelli”. Patients provided informed consent for longitudinal plasma collection and tumor DNA profiling (Protocol n° 59, approved by the Ethics Committee of the University of Campania “Luigi Vanvitelli”, in accordance with the Declaration of Helsinki). Sum of lesion diameters (SoD) was calculated as the sum of the maximum diameters, expressed in millimeters, of all measurable lesions from whole-body CT scans. Tumor response was assessed using Response Evaluation Criteria in Solid Tumours (RECIST) v1.1. Lactate dehydrogenase (LDH) levels for the correlation analysis was expressed as a ratio (LDH value/upper limit of normal LDH as per internal laboratory reference values). Performance status (PS) was assessed using the Eastern Cooperative Oncology Group (ECOG) scale. Data cut-off for survival analysis was 31 October 2021.

### 2.2. Tissue Analysis

Analyses of the formalin-fixed, paraffin-embedded (FFPE) tissue specimens were all performed in the Pathology Service of University of Campania “Luigi Vanvitelli” using Next Generation Sequencing (NGS) as described in Appendix A.

### 2.3. Plasma Collection and qPCR Analysis

Plasma was collected and analyzed immediately after centrifugation or stored at −80 °C until analysis, as described in the Appendix A. Analyses of plasma were all carried out using the automated Idylla™ qPCR-based platform by Biocartis (Mechelen, Belgium), as previously described (Appendix A) [21,22]. The maximum waiting time from collection to centrifugation of the samples was less than 1 h.

### 2.4. Statistical Analysis

After checking the assumptions of a normal distribution of the values for the quantitation cycle (Cq), circulating mutational fraction (CMF, %), SoD (mm), and LDH (ratio), using the Anderson–Darling test (normal distribution for Cq and SoD, non-normal distribution for CMF and LDH), a Spearman test was used to assess the pairwise correlation between Cq and both LDH and SoD, and also between CMF and both LDH and SoD. Outlier values were identified using the ROUT (Q = 1%) method [23]. The Kaplan–Meier method was used for survival analysis, and the significance of the split between the survival curves were measured by the log-rank (Mantel–Cox) test. All statistical analyses were performed using GraphPad Prism 8.0.1 software.(GraphPad Software Inc, San Diego, CA, USA)

## 3. Results

### 3.1. Patients’ Characteristics

Among the 56 stage III/IV melanoma patients, 30 had unresected disease at the time of first plasma collection (advanced cohort) and 26 had undergone radical resection (confirmed both on pathologic and radiologic assessment) (resected cohort) within 3 months before the first plasma collection. In particular, among the 30 patients in the advanced cohort, 25 were assessed before starting any systemic treatment (baseline cases) and 5 were assessed during systemic treatment (non-baseline cases). Moreover, for 10 out of the 25 baseline patients, the plasma sample was collected also at the time of progression disease or after 6 months in the absence of disease progression (Figure 1).

Patients’ characteristics are summarized in Table 1 and Table 2.

### 3.2. Baseline Locally Advanced and Metastatic Patients (Baseline Cohort)

Among the 25 consecutive treatment-naive BRAF-V600/NRAS mutant patients with locally advanced or metastatic melanoma who were tested at baseline, the same mutation of the tissues was found in the plasma in 15 patients, with a global concordance of 60%. We therefore analyzed patients’ baseline clinical characteristics and correlate them to qPCR results. In particular, we focused on tumor burden, metastatic sites, LDH levels, and disease-related symptoms (using ECOG PS scale) (Table 3).

Interestingly, plasma analysis was positive in all BRAF-V600 mutant patients, with SoD ≥ 30 mm (13/13); extrapulmonary visceral, excluding exclusive brain metastases (8/8), with ≥2 metastatic sites (8/8), with liver metastases (6/6), with elevated baseline LDH levels (6/6), or with symptomatic disease (4/4). Conversely, ctDNA analysis did not reveal blood mutations in all three patients with locally advanced disease and in the two patients with CNS-limited disease. M1a stage patients were found positive in 7 out of 13 patients (53.8%), but, in case of nodal metastases and SoD ≥ 30 mm, the positivity rate was higher (87.5%). The miss rate (or false negative rate, FNR) was calculated for the baseline cohort, this being 0.4. The characteristics most closely related to false negative results were low SoD (<30 mm) and cutaneous or nodal disease only.

After a median follow-up of 18.7 months (range 3.5–39.7), 17 (68%) patients had progressed to first-line treatment and 14 (56%) had died. The median progression-free survival (PFS) and overall survival (OS) was 13.3 and 18.7 months, respectively.

PFS was calculated for baseline patients from the time of first plasma collection to time of disease progression or death of any cause, whichever occurred first; similarly, OS was calculated for baseline patients from the time of first plasma collection to death of any cause. For both PFS and OS analysis, we divided the baseline patients in positive and negative groups according to their ctDNA result (Figure 2).

A non-statistically significant difference between the negative and positive groups was observed for both PFS (median PFS: 13.8 vs. 12.4 months, respectively; HR: 0.85, 95%CI: 0.33–2.2, *p* = 0.74) and OS (median OS: 25.2 vs. 21.1 months, respectively; HR: 0.71, 95%CI: 0.25–2.02, *p* = 0.39) (Figure 2).

### 3.3. Non-Baseline Locally Advanced and Metastatic Patients

Four BRAF-V600/NRAS mutant pre-treated patients with locally advanced or metastatic melanoma and one patient (#A30) with de novo metastases during adjuvant therapy for radically resected BRAF-V600/NRAS wild-type melanoma were tested. Only 1 case out of 4 was ctDNA positive for the BRAF-V600/NRAS mutation (Table 4).

Notably, patient #A30 was in treatment with anti-PD-1 adjuvant therapy for radically resected stage IIIC melanoma; the BRAF-V600/NRAS analysis was performed on the metastatic sentinel node, and no mutation was found on this tissue. However, for rapid development of symptomatic metastases (bone, liver, and lungs, SoD: 70 mm, elevated LDH level) at 6 months after starting adjuvant treatment, we tested a plasma sample and, surprisingly, a BRAF-V600 mutation was found. The patient underwent liver biopsy, which confirmed the presence of the BRAF-V600 mutation, allowing for the initiation of BRAF and MEK inhibitors.

### 3.4. Monitoring of Locally Advanced and Metastatic BRAF-V600 Patients

Ten BRAF-V600 mutant locally advanced and metastatic patients repeated plasma sample collection after 6 months, if their plasma sample was positive at baseline, or at evidence of progressive disease (PD), in case their plasma sample was negative at baseline (Table 5). 

Of eight patients with a mutation detected in their baseline plasma sample, only one patient was still positive after 6 months (#A11); notably, it was also the only patient with concurrent progressive disease at CT scan. Of the two patients with a negative baseline plasma sample, a new analysis was performed at the time of PD: patient #A7, with de-novo liver metastases (SoD: 15 mm) after 10 months of targeted therapy, was found positive (same mutation of primary tumor), but patient #A10, despite evidence of nodal PD (SoD: 50 mm) after 12 months of targeted therapy, was still negative; intriguingly, a new molecular analysis on nodal biopsy at time of PD in patient #A10 was performed, and no BRAF-V600/NRAS mutation was found.

### 3.5. Cq and CMF in Advanced Patients’ Cohort

Cq values are reported in Appendix A, showing a normal distribution (Appendix A); the median Cq value in baseline cohort was 49.46 (range: 36.94–53.4). Therefore, we obtained PFS and OS curves dividing patients in two groups according to their Cq values (< or ≥49.46), including negative patients in the group with higher Cq values—starting from the assumption that higher Cq values mean a lower quantity of ctDNA and vice versa (Figure 2). PFS analysis showed a non-statistically significant difference between the Cq low and Cq high groups (median PFS: 15.5 vs. 6.9 months, respectively; HR: 0.46, 95%CI: 0.14–1.51, *p* = 0.12) while a difference in OS was observed (median OS: 25.3 vs. 10.7 months, respectively; HR: 0.32, 95%CI: 0.09–1.21, *p* = 0.027) (Figure 3).

CMF values are reported in Appendix A; a non-normal distribution of all CMF values was observed (Appendix A). In a similar way to Cq, we obtained PFS and OS curves according to the CMF values (< or ≥0.011%, which is the median value of CMF in the baseline cohort excluding the two outlier values), including negative patients in the group with lower CMF values (Figure 2). The result was not significant for the correlation of CMF with both PFS and OS (Appendix A). Moreover, we analyzed the potential correlation between the Cq and CMF values with clinical factors such as the SoD and LDH ratios, finding no correlation (Appendix A).

### 3.6. Baseline Radically Resected Stage III–IV Patients (Resected Cohort)

Among the 26 consecutive radically resected stage III–IV melanoma patients who were tested before starting adjuvant treatment, the BRAF V600E mutation was found on ctDNA only in one patient. At the data cut-off time, after a median follow up of 26.7 months (range: 14.3–38.9), 9 out of 26 patients (34.6%) had disease relapse, with a median disease-free survival (DFS, defined as the time from randomization to recurrence of tumor or death whichever occurred first) of 20.6 months (range: 11.6–38.9) (Table 6).

Noteworthy, among the relapsed patients, there was the only one of this cohort with a positive ctDNA result at baseline (#B6, see Table 6), who developed CNS metastases at the end of the year of adjuvant treatment with targeted therapy.

Accuracy in relapse detection by qPCR on ctDNA was therefore calculated in this cohort: the positive predictive value (PPV) was 100% and negative predictive value (NPV) was 68%, with an FNR of 0.875, reflecting an extremely low power for identifying patients at higher risk of relapse.

## 4. Discussion

The present work has evaluated the overall clinical performance of Idylla™ Biocartis in the characterization of BRAF/NRAS-mutated melanoma patients, either in radically operated stage III–IV or in locally advanced/metastatic ones.

Concerning locally advanced and metastatic melanoma patients (advanced cohort), we obtained an overall plasma–tissue concordance of 60%, despite a lower sensitivity compared to other similar methods. The same technology had been already investigated in metastatic melanoma patients in two previous works [13,14], reporting baseline plasma–tissue concordance for the BRAF mutation of 47% and 64.2%, respectively, whilst an overall agreement of 84% was shown in the work by Long-Mira et al. [11]. Moreover, we investigated the correlation between ctDNA quantity and clinical–radiological tumor parameters, in our baseline cohort. In particular, a high rate of circulating mutation identification was obtained in patients with high burden of disease, high LDH levels, and/or symptomatic disease; all these characteristics are in fact associated with the highest probability of finding relevant ctDNA concentration in plasma samples, detecting BRAF-V600/NRAS mutations virtually in all cases. On the contrary, we did not find any mutation in plasma samples from metastatic patients with the brain as the unique site of metastasis; this finding, in line with previous reports [13,24,25,26], is probably linked to a lower ctDNA quantity released into the circulation by the blood–brain barrier; similarly, patients with locally advanced disease likely have ctDNA levels lower than the sensibility threshold of this technique.

With respect to the prognostic value of ctDNA, previous works highlighted a correlation between ctDNA levels and survival [13,15,16,25]. In our work, we used Cq values to investigate their potential prognostic significance, finding a non-significant trend towards a better PFS and a statistically significant improvement in OS in those patients without detectable mutations and with a Cq value ≥49.46 (median value in test-positive patients). Similarly, Rutkowsky and colleagues failed to demonstrate a correlation between Cq values and PFS, and also between Cq and duration of response (DoR) [14].

In line with previous reports [11,13,25], our results support the potential use of liquid biopsy to monitor the response to treatment together with radiological and/or clinical assessment in locally advanced or metastatic melanoma patients with a BRAF-V600/NRAS mutation already identified by baseline liquid biopsy. The interesting cases of patient #A10, in which ctDNA remained negative despite disease progression to iliac lymph nodes—a result which was later confirmed to be truly BRAF WT at tissue analysis—highlight a good clinical correlation that better recapitulate melanoma biology in a clinical scenario. Moreover, in patient #A30, BRAF-V600 ctDNA positivity constituted a de novo event during anti-PD1 adjuvant treatment and, after confirmation on metastatic tissue analysis, allowed the patient to access targeted therapy. In this circumstance, ctDNA analysis was used to better capture tumor heterogeneity and allowed the patient to access alternative treatment strategies in a short period of time.

Finally, concerning radically resected stage III/IV melanoma patients, our results indicate that qPCR-based plasma analysis could not be used in predicting disease relapses. In fact, despite the high PPV, the low sensitivity of the test in this scenario translates into a high FNR. In this setting, more sensitive techniques, such as ddPCR, better correlate with relapse, as previously described [20].

Taken all together, our data suggest that qPCR-based ctDNA analysis on plasma samples using Idylla™ Biocartis (Mechelen, Belgium) could be used to achieve a better understanding of melanoma biology and provides valuable clinical information in patients with specific clinical–radiological characteristics, in addition to the current gold-standard tissue-based mutational analysis [26]. In particular, the performance of this technique in disease monitoring for advanced disease is worth further investigation and validation, while its potential for the identification of relapsing patients is not clinically reliable to differentiate patients at higher risk of relapse, though the presence of detectable ctDNA is strongly associated with relapse before imaging detection.

Among the advantages of this method, it must be underlined that the time required to obtain the analysis is approximatively 120 min after sampling, possibly allowing to anticipate access to targeted patients, if our results are prospectively confirmed in larger cohorts, especially for symptomatic patients with a high disease burden who could benefit from the rapid effect of targeted therapy without delay [27]. Limitations of our work are the inclusion of a low number of patients with a known BRAF-V600/NRAS mutation in tissues and the impossibility of excluding confounding factors that limit our analysis, particularly for the Cq and CMF values.

Finally, with regard to cost-effectiveness assessments that are strictly dependent on the healthcare system of reference, a definitive estimation cannot be accurately derived from the present study. However, the identification of a target population with higher diagnostic accuracy of the test achieved by this study can strongly enhance its feasibility and effectiveness by refining patients’ selection, and this technique was recently shown to be the cheapest in centers with a low sample throughput per year [28].

## 5. Conclusions

Our work shows the potential clinical utility of a ready-to-use diagnostic tool in stage III–IV melanoma patients, from molecular diagnosis to response monitoring, whose results could be integrated with the currently used clinical–radiological factors. Results from our work, if prospectively validated using a wider cohort of patients, could therefore improve the outcomes of melanoma patients. In fact, the automated qPCR-based ctDNA analysis using this platform could provide useful information in a very short timeframe and help decision making for the treating clinicians.

## Figures and Tables

**Figure 1 cancers-14-03053-f001:**
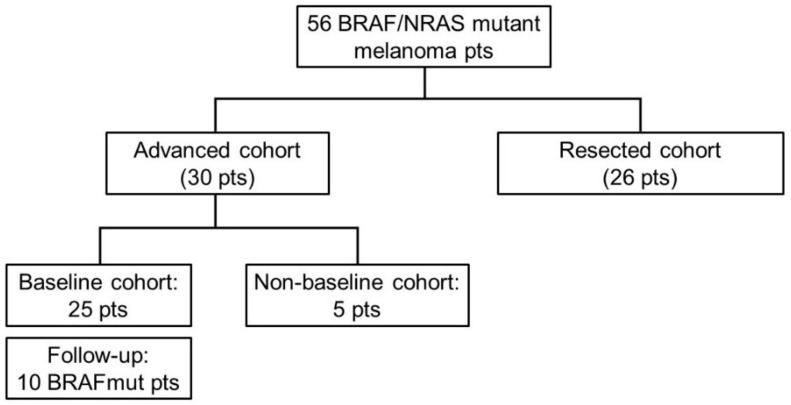
Flowchart of the enrolled patients. Pts: patients.

**Figure 2 cancers-14-03053-f002:**
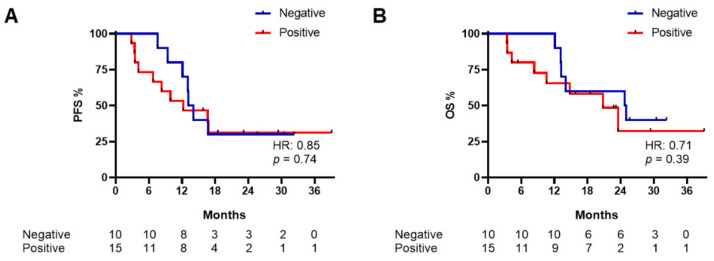
Progression-free survival (PFS) (**A**) and overall survival (OS) (**B**) in baseline patients according to their qPCR result.

**Figure 3 cancers-14-03053-f003:**
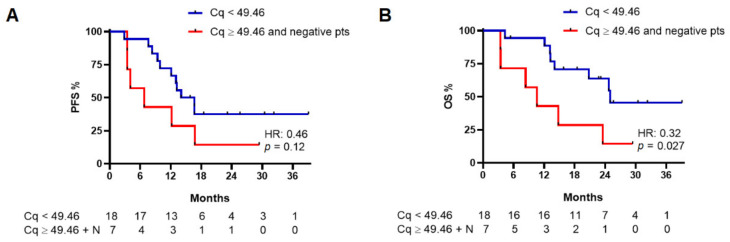
Progression-free survival (PFS) (**A**) and overall survival (OS) (**B**) in baseline patients with a Cq value lower than 49.46 and patients with a Cq value equal to or higher than 49.46 (including in this group patients with no mutation detected by the instrument; see text for details). N, patients with no detectable mutation at baseline.

**Table 1 cancers-14-03053-t001:** Characteristics of the locally advanced/metastatic patients (advanced cohort).

Characteristic	Locally Advanced/Metastatic Patients (30)
Age: median (range)	62 years (34–86)
Sex	
- Male	16 (53.3%)
- Female	14 (46.7%)
Stage (AJCC VIII ed)	
- Locally advanced	3 (10%)
- M1a	14 (46.7%)
- M1b	0 (0%)
- M1c	9 (30%)
- M1d	4 (13.3%)
Metastatic site	
- Skin	10 (33.3%)
- Node	14 (46.7%)
- Liver	6 (20%)
- Lung	3 (10%)
- Brain	3 (10%)
- Other	5 (16.6%)
- ≥2 sites	8 (26.7%)
SoD: median (range)	45 mm (10–125)
Mutation (tissue analysis)	
- BRAF V600 mut	24 (80%)
V600E	18 (60%)
V600K	4 (13.3%)
V600R	1 (3.3%)
- NRAS mut	6 (20%)
Exon 3	5 (16.6%)
Exon 2	1 (3.3%)
LDH	
- Normal	24 (80%)
- High (>ULN)	6 (20%)
ECOG PS	
- 0	26 (86.6%)
- 1	4 (13.3%)
First line therapy	
- Immunotherapy	10 (33.3%)
- Targeted therapy	20 (66.7%)

Abbreviations: SoD, sum of diameters of lesions; LDH, lactate dehydrogenase; ECOG PS, Eastern Cooperative Oncology Group Performance Status scale.

**Table 2 cancers-14-03053-t002:** Characteristics of the radically resected stage III/IV patients (resected cohort).

Characteristic	Radically Resected Stage III/IV Patients (26)
Age: median (range)	52 years (20–81)
Sex	
- Male	17 (65.4%)
- Female	9 (34.6%)
Stage (AJCC VIII ed)	
- IIIA	1 (3.8%)
- IIIB	8 (30.8%)
- IIIC	15 (57.7%)
- IIID	0 (0%)
- IV	2 (7.7%)
Mutation (tissue analysis)	
- BRAF V600 mut	20 (76.9%)
V600E	17 (65.4%)
V600K	3 (11.5%)
- NRAS mut	6 (23.1%)
Exon 3	6 (23.1%)
Adjuvant therapy	
- Immunotherapy	9 (34.6%)
- Targeted therapy	17 (65.4%)

Abbreviations: AJCC, American Joint Committee on Cancer.

**Table 3 cancers-14-03053-t003:** Baseline locally advanced and metastatic patients.

Patients’ Characteristics												Sites of Disease						
Pt n°	Age	Sex	Mutation on FFPE/FNA	Specimen	Baseline qPCR Analysis Results	Cq	CMF (%)	AJCC 8th	LDH Level	ECOG PS	SoD (mm)	Skin	Nodes	Liver	Lung	Brain	Other Sites	≥2 Sites
A1	62	F	BRAF V600E	Liver metastases	BRAF V600E/D	44.73	0.237	M1c	H	0	60	-	-	Yes	-	-	-	-
A2	67	M	BRAF V600K	Primary	BRAF V600K-R	52.96	0.004	M1a	N	0	50	Yes	Yes	-	-	-	-	Yes
A3	86	M	BRAF V600R	Primary	BRAF V600K-R	45.16	0.111	M1c	H	0	70	-	Yes	Yes	-	-	Yes	Yes
A4	50	F	BRAF V600E	Lymph node	BRAF V600E-D	44.98	0.234	M1a	N	1	46	-	Yes	-	-	-	-	-
A5	53	F	BRAF V600E	Primary	BRAF V600E-D	52.65	0.008	M1c	H	0	85	Yes	Yes	-	-	-	Yes	Yes
A6	71	F	NRAS Q61K	Primary	NRAS Q61R-K	53.13	0.0004	M1a	N	1	97	-	Yes	-	-	-	-	-
A7	79	F	BRAF V600K	Primary	negative	X	X	L.A.	N	0	<10	Yes	-	-	-	-	-	-
A8	73	M	NRAS Q61R	Primary	negative	X	X	M1a	N	0	67	Yes	-	-	-	-	-	-
A9	57	M	BRAF V600E	Primary	negative	X	X	M1a	N	0	20	Yes	-	-	-	-	-	-
A10	40	F	BRAF V600E	Lymph node	negative	X	X	M1a	N	0	13	-	Yes	-	-	-	-	-
A11	59	F	BRAF V600K	Primary	BRAF V600K-R	47.2	0.249	M1c	H	0	45	-	-	Yes	-	-	-	-
A12	66	F	BRAF V600E	Primary	negative	X	X	M1a	N	0	10	Yes	-	-	-	-	-	-
A13	48	M	BRAF V600E	Primary	BRAF V600E-D	41.29	5.441	M1c	N	0	65	-	Yes	-	-	-	Yes	Yes
A14	50	F	BRAF V600E	Primary	negative	X	X	M1d	N	0	15	-	-	-	-	Yes	-	-
A15	73	F	NRAS Q61R	Primary	NRAS Q61R-K	44.95	0.051	M1c	N	0	30	-	-	Yes	Yes	-	-	Yes
A16	79	F	NRAS Q61R	Primary	negative	X	X	L.A.	N	0	<10	Yes	-	-	-	-	-	-
A17	54	M	BRAF V600K	Primary	BRAF V600K-R	50.08	0.034	M1c	N	0	125	Yes	Yes	Yes	Yes	-	Yes	Yes
A18	75	M	BRAF V600E	Lymph node	BRAF V600E-D	53.4	0.002	M1a	N	0	42	-	Yes	-	-	-	-	-
A19	78	F	NRAS G12C	Primary	negative	X	X	M1a	N	0	46	-	Yes	-	-	-	-	-
A20	62	M	BRAF V600E	Lymph node	negative	X	X	M1d	N	0	10	-	-	-	-	Yes	-	-
A21	59	M	BRAF V600E	Lymph node	BRAF V600E-D	36.94	0.639	M1c	H	1	75	-	Yes	Yes	Yes	Yes	Yes	Yes
A22	64	F	BRAF V600E	Lymph node	BRAF V600E-D	51.17	0.011	M1a	N	0	40	-	Yes	-	-	-	-	-
A23	63	M	BRAF V600E	Lymph node	BRAF V600E-D	49.46	0.0003	M1a	H	1	38	-	Yes	-	-	-	-	-
A24	54	M	BRAF V600E	Lymph node	BRAF V600E-D	51.05	0.0001	M1a	N	0	10	Yes	Yes	-	-	-	-	Yes
A25	84	F	BRAF V600E	Skin metastasis	negative	X	X	M1a	N	0	12	Yes	-	-	-	-	-	-

AJCC: American Joint Committee on Cancer; CMF, circulating mutational fraction; Cq, quantitation cycle; ECOG PS, Eastern Cooperative Oncology Group performance status; FFPE, formalin-fixed paraffin-embedded; FNA, fine needle aspiration; H, higher than the upper limit of normality; N, within normal range; qPCR, quantitative PCR; SoD, sum of diameters.

**Table 4 cancers-14-03053-t004:** Non-baseline locally advanced and metastatic patients.

Pt n°	Age	Sex	Mutation on FFPE-FNA	Specimen	Baseline Stage AJCC 8th	qPCR Analysis Results	Sites of Disease at Time of Analysis	SoD (mm) at Time of Analysis	Clinical Information at the Time of Biocartis Analysis
A26	42	M	NRAS Q61R	Lymph node	M1a	Negative	N	15	Low tumor burden at baseline During treatment (anti-PD-1): PR
A27	35	M	BRAF V600E	Lymph node	M1c	negative	Li, Lu, N	60	High tumor burden at baseline During treatment (TT): PR
A28	64	M	BRAF V600E	Primary tumor	L.A.	negative	Sk, N	54	Locally advanced at baseline During treatment (TT): PR
A29	42	M	BRAF V600E	Brain metastases	M1d	BRAF V600E-D	Lu, N	45	High tumor burden at baseline During treatment (anti-PD-1): PR
A30	34	M	BRAF wt (BRAF V600E) *	Lymph node (liver metastases)	IIIC Radically resected	BRAF V600E-D	Li, Lu, Bo	70	De novo symptomatic metastases, 6 months after starting adjuvant therapy Elevated LDH

AJCC, American Joint Committee on Cancer; Bo, bones; FFPE, formalin-fixed paraffin-embedded; FNA, fine needle aspiration; L.A., locally advanced; Li, liver; N, nodes; PR, partial response; qPCR, quantitative PCR; Sk: skin; SoD, sum of diameters; TT, targeted therapy; wt, wild type. * Discordance between node and liver metastases: biopsy on liver metastases was performed after liquid biopsy result to confirm the presence of the BRAF-V600 mutation.

**Table 5 cancers-14-03053-t005:** Monitoring of BRAF-V600 mutant locally advanced and metastatic patients after 6 months of treatment.

Pt n°	Baseline Stage AJCC 8th	Sites of Disease at Baseline	Baseline SoD (mm)	Baseline qPCR Analysis Results	First-Line Therapy	Response (RECIST)	Sites at Second Analysis	Second SoD (mm)	Second qPCR Analysis Results
A2	M1a	Sk, N	50	BRAF V600K-R	TT	CR	None	0	negative
A4	M1a	N	46	BRAF V600E-D	TT	PR	N	14	negative
A5	M1c	Ad, Sk, N	85	BRAF V600E-D	TT	SD	Ad, Sk, N	80	negative
A7	L.A.	Sk	10	0	TT	PD	Li	15	BRAF V600K-R
A10	M1a	N	13	0	TT	PD	N	50	negative
A11	M1c	Li	45	BRAF V600K-R	TT	PD	Li, Lu	90	BRAF V600K-R
A13	M1c	N, Pe	65	BRAF V600E-D	TT	CR	None	0	negative
A17	M1c	Li, Sp, Sk, N	125	BRAF V600K-R	Anti-PD-1	PR	Li, Sp, N	30	negative
A18	M1a	N	42	BRAF V600E-D	TT	PR	N	24	negative
A30	M1c	Li, Lu, Bo	70	BRAF V600E-D	TT	PR	Li, Bo	40	negative

Ad: adrenal glands; AJCC, American Joint Committee on Cancer; Bo, bones; CR, complete response; L.A., locally advanced; Li, liver; N, nodes; PD, progressive disease; Pe, peritoneum; PR, partial response; qPCR, quantitative PCR; SD, stable disease; Sk, skin; SoD, sum of diameters; Sp, spleen; TT, targeted therapy.

**Table 6 cancers-14-03053-t006:** Radically resected stage III–IV patients.

Patients’ Characteristics							
Pt n°	Age	Sex	Mutation on FFPE-FNA	Stage AJCC 8th	Baseline qPCR Analysis Results	Adjuvant Therapy	Relapse (If Yes, Which Sites)
B1	55	F	BRAF V600E	IIIB	negative	TT	-
B2	48	M	BRAF V600E	IIIB	negative	TT	-
B3	41	F	NRAS Q61R	IIIC	negative	Anti-PD-1	-
B4	48	F	NRAS Q61R	IIIC	negative	Anti-PD-1	Yes, brain
B5	52	M	BRAF V600K	IIIB	negative	TT	-
B6	54	F	BRAF V600E	IIIC	BRAF V600E-D	TT	Yes, brain
B7	41	M	BRAF V600E	IIIA	negative	TT	-
B8	78	M	BRAF V600K	IIIC	negative	Anti-PD-1	-
B9	81	M	NRAS Q61R	IIIC	negative	Anti-PD-1	Yes, loco-regional
B10	49	F	BRAF V600E	IIIC	negative	TT	Yes, skin
B11	52	M	BRAF V600E	IIIC	negative	TT	-
B12	37	F	BRAF V600E	IIIC	negative	TT	Yes, skin
B13	53	M	BRAF V600E	IIIC	negative	Anti-PD-1	Yes, skin
B14	47	M	BRAF V600E	IIIC	negative	TT	-
B15	35	M	BRAF V600E	IIIB	negative	TT	-
B16	43	F	BRAF V600E	IIIC	negative	TT	-
B17	59	M	BRAF V600E	IIIC	negative	TT	Yes, liver and spleen
B18	39	F	BRAF V600E	IIIC	negative	TT	Yes, nodal
B19	20	M	BRAF V600E	IIIB	negative	TT	-
B20	62	M	NRAS Q61R	IIIC	negative	Anti-PD-1	Yes, lung
B21	75	F	NRAS Q61L	IV R0	negative	Anti-PD-1	-
B22	73	M	NRAS Q61R	IV R0	negative	Anti-PD-1	-
B23	63	M	BRAF V600E	IIIB	negative	TT	-
B24	65	M	BRAF V600E	IIIB	negative	TT	-
B25	76	M	BRAF V600K	IIIC	negative	Anti-PD-1	-
B26	39	M	BRAF V600E	IIIB	negative	TT	-

AJCC, American Joint Committee on Cancer; FFPE, formalin-fixed paraffin-embedded; FNA, fine needle aspiration; qPCR, quantitative PCR; R0, radically resected; TT, targeted therapy.

## Data Availability

The data presented in this study are available in this article (and Appendix A).

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
