# Peer review of "Clinical Utility of Liquid Biopsy to Detect BRAF and NRAS Mutations in Stage III/IV Melanoma Patients by Using Real-Time PCR"

_cancers, 2022, doi:10.3390/cancers14133053_

Round 1

Reviewer 1 Report

In the present study, the value of qPCR-based liquid biopsy for the detection of therapeutically relevant mutations in patients with stage III/IV melanoma was investigated. The result of sequencing tissue samples was used as the gold standard. The liquid biopsy results of a total of 56 patients were evaluated, of whom approx. 50 % had already been resected at the time of the initial examination.

The results show that liquid biopsy is only able to achieve results in the sense of the research question if the tumour burden of the individual patient is already very high at the time of examination.

Prediction of relapse: The calculation of PPV and NPV is not statistically reasonable if you only have one true positive patient in the investigation! Otherwise, you would also have to mention the sensitivity of 11 %.

Discussion: The discussion correctly describes that there will probably only be a small number of patients who will benefit from the relatively insensitive methodology. Accordingly, it should be discussed whether the gain in information can justify the acquisition of the corresponding technology. Please check your references: P. 13 “[Tan]”?

Author Response

In the present study, the value of qPCR-based liquid biopsy for the detection of therapeutically relevant mutations in patients with stage III/IV melanoma was investigated. The result of sequencing tissue samples was used as the gold standard. The liquid biopsy results of a total of 56 patients were evaluated, of whom approx. 50 % had already been resected at the time of the initial examination.

- The results show that liquid biopsy is only able to achieve results in the sense of the research question if the tumour burden of the individual patient is already very high at the time of examination.

We wish to thank the Reviewer for this comment that we completely agree with. It is true that the results we show can be mainly applicable to those cases when the melanoma tumor burden is high, but this peculiarity can be exploited to identify a clinically recognizable subset of patients for which PCR-based liquid biopsy can be used with acceptable clinical significance in terms of disease monitoring.

- Prediction of relapse: The calculation of PPV and NPV is not statistically reasonable if you only have one true positive patient in the investigation! Otherwise, you would also have to mention the sensitivity of 11 %.

We thank the Reviewer for this comment. The diagnostic power of our approach in identifying relapse is definitively low, as a high enough tumor burden is needed to have a reliable test result. The sensitivity is too low to propose this test for this purpose, and the only clear conclusion that can be drawn is that a positive ctDNA result after surgical resection is strongly predictive of recurrence, while the other way around is not true. We updated the manuscript for clarity

- Discussion: The discussion correctly describes that there will probably only be a small number of patients who will benefit from the relatively insensitive methodology. Accordingly, it should be discussed whether the gain in information can justify the acquisition of the corresponding technology.

We thank you for pointing out this issue. Our work is outside of cost-effectiveness assessments. Until now, there have been few studies assessing the financial costs of this platform against the standard of care. An adequate estimation of this parameter would include a greater number of patients and would consider specific health-related costs depending on the healthcare system of reference. However, we can make some considerations: liquid biopsy, as we suggested, can be carried out in the first instance only when treatment is required in a very short time in symptomatic patients with a high burden of disease: in this way, the test is only an implementation of the normal SOC and not an addition; if we consider only the cost of a single cartridge compared to the reimbursement of an NGS in our region, they are very similar; finally, a recent analysis showed that among the most common liquid biopsy techniques, this one we use is the cheapest at low sample throughput per year (<110 samples) (Performance of four platforms for KRAS mutation detection in plasma cell-free DNA: ddPCR, Idylla, COBAS z480 and BEAMing, Vessies et al., Scientific Reports, 2020). These considerations have been added to the text.

- Please check your references: P. 13 “[Tan]”?

We thank the reviewer for having noticed it, we modified it with the correct reference number.

Reviewer 2 Report

Although the current article by Giunta et al., Clinical utility of liquid biopsy to detect BRAF and NRAS mutations, is written in a professional manner, it tries to demonstrate the potential clinical utility of a ready-to-use diagnostic tool in stage III-IV melanoma patients, from molecular diagnosis to response monitoring, whose results could be integrated with the currently used clinical-radiological factors. As the ctDNA approach becomes more widely used in cancer diagnosis, it must be thoroughly standardized and confirmed before concluding the results, as well as before being integrated into clinical practice. I don't believe this article can be published in its current state; it has to be revised with advance arguments. Here are some recommendations that may assist improve the quality of the current manuscript:-

- How did the author exclude the possibility of a false-negative rate using a liquid biopsy-based RT-PCR technique?

-Given that a liquid biopsy does not substitute a tissue-based diagnosis, how can the authors be certain that it will aid clinical utility?

- The cost of these approaches should be compared to tissue-based diagnosis by the author.

- The author must describe the entire Plasms collection process used in this study, as it is critical to reduce the time between drawing blood and separating plasma or serum from other blood elements, or using specialized cell stabilizing media, to avoid artifacts such as ctDNA dilution by non-tumor genomic DNA or nuclease degradation, which can lead to unreliable results.

- The sample size is too small, especially with the known BRAF-V600/NRAS mutation, hence the sample size must be increased in order to draw any conclusions.

- Is the author aware of sample collection time kinetics, that is, how do the data look following a second collection of the sample from the same patient after 48 or 72 hours?

- More information about the sample selection criteria for the current study's inclusion and exclusion criteria is needed.

Author Response

Although the current article by Giunta et al., Clinical utility of liquid biopsy to detect BRAF and NRAS mutations, is written in a professional manner, it tries to demonstrate the potential clinical utility of a ready-to-use diagnostic tool in stage III-IV melanoma patients, from molecular diagnosis to response monitoring, whose results could be integrated with the currently used clinical-radiological factors. As the ctDNA approach becomes more widely used in cancer diagnosis, it must be thoroughly standardized and confirmed before concluding the results, as well as before being integrated into clinical practice. I don't believe this article can be published in its current state; it has to be revised with advance arguments. Here are some recommendations that may assist improve the quality of the current manuscript:

- How did the author exclude the possibility of a false-negative rate using a liquid biopsy-based RT-PCR technique

We thank the reviewer for this comment. Of course, we did not exclude the possibility of false negative results given the inherently low sensitivity of the technique. Concerning locally advanced and metastatic cohort, we used the NGS test results as the gold standard comparator: we’ve added FNR for this cohort in the text, accordingly (paragraph 3.2). On the other hand, concerning radically resected melanoma patients, our aim was to test the sensitivity of the RT-PCR technique in a specific setting where few positive cases were expected: therefore, FNR was not calculated.

- Given that a liquid biopsy does not substitute a tissue-based diagnosis, how can the authors be certain that it will aid clinical utility?

We wish to thank the Reviewer for this comment. The main objective of our work is to determine the diagnostic and prognostic potential of this easy-to-use automated platform for ctDNA analysis in BRAF/NRAS mutated melanoma, and the clinico-pathological features associated with higher or lower detection power. We agree that, as it is now, this technique cannot substitute tissue-based molecular analyses, but we believe that it can still be informative of the tumor biology and the clinical behavior of the disease. In particular, in case of advanced disease, the Cq value for BRAF/NRAS mutation in ctDNA presents a statistically significant prognostic meaning (OS), and a negative ctDNA analysis during therapy is associated with clinical response; on the other hand, in case of clinically totally resected melanoma, the presence of detectable ctDNA is definitively associated with relapse before imaging detection.
Clearly, our data must be interpreted with caution as they derive from a single center case series and are thus subject to biases, but we still believe that the clinical relevance of a more insightful understanding of the biology of the disease deriving from our work can be meaningful. We implemented the discussion for clarity

- The cost of these approaches should be compared to tissue-based diagnosis by the author.

We thank you for pointing out this issue. Our work is outside of cost-effectiveness assessments. Until now, there have been few studies assessing the financial costs of this platform against the standard of care. An adequate estimation of this parameter would include a greater number of patients and would consider specific health-related costs depending on the healthcare system of reference. However, we can make some considerations: liquid biopsy, as we suggested, can be carried out in the first instance only when treatment is required in a very short time in symptomatic patients with a high burden of disease: in this way, the test is only an implementation of the normal SOC and not an addition; if we consider only the cost of a single cartridge compared to the reimbursement of an NGS in our region, they are very similar; finally, a recent analysis showed that among the most common liquid biopsy techniques, this one we use is the cheapest at low sample throughput per year (<110 samples) (Performance of four platforms for KRAS mutation detection in plasma cell-free DNA: ddPCR, Idylla, COBAS z480 and BEAMing, Vessies et al., Scientific Reports, 2020). These considerations have been added to the text.

- The author must describe the entire Plasms collection process used in this study, as it is critical to reduce the time between drawing blood and separating plasma or serum from other blood elements, or using specialized cell stabilizing media, to avoid artifacts such as ctDNA dilution by non-tumor genomic DNA or nuclease degradation, which can lead to unreliable results.

Thanks for the request for precisation. To avoid the phenomenon of nuclease degradation or other biases, we processed the samples immediately after collection. The maximum waiting time from collection to centrifugation was less than 1 hour. We have added this detail to the text.

- The sample size is too small, especially with the known BRAF-V600/NRAS mutation, hence the sample size must be increased in order to draw any conclusions.

We wish to thank the Reviewer for this suggestion. It would be indeed desirable to have a larger sample size in order to increase the statistical significance of the results. However, we designed this study with the intent to explore the potential of this automated ctDNA platform, characterized by easy sample processing albeit a relatively low sensitivity, in characterizing melanoma patients with different clinico-pathological features. For this reason, this study must be considered as exploratory and instrumental to identify the best clinical settings to investigate the use of this technology in larger patients’ cohorts. We added this considerations to the discussion section of the paper.

- Is the author aware of sample collection time kinetics, that is, how do the data look following a second collection of the sample from the same patient after 48 or 72 hours?

We wish to thank the Reviewer for this insightful comment of ctDNA kinetics. Unfortunately, we did not collect multiple samples for the same patients at short intervals (48-72 hrs), since the results of the test were not used interventionally and the main goal of the work was not to validate its sensitivity at different timepoints. Conversely, we designed the collection in order to correlate specific clinico-pathological features of melanoma patients with the positivity of ctDNA for BRAF/NRAS mutations in different pre-specified clinical settings with a single blood collection.

- More information about the sample selection criteria for the current study's inclusion and exclusion criteria is needed.

We thank the reviewer, we added a new paragraph in the supplementary method section, listing inclusion and exclusion criteria for each of the three cohorts.

Round 2

Reviewer 2 Report

Thank you for responding. It appears that the author has satisfactorily addressed the majority of the questions, and no further changes are required. The manuscript is now in good shape, and it can be accepted in its current form without any further changes.

Thanks